# Effect of No-Tillage Management on Soil Organic Matter and Net Greenhouse Gas Fluxes in a Rice-Oilseed Rape Cropping System

Huabin Zheng [1] , Xianliang Tang [1], Jiabin Wei [1], Huaqin Xu [2,*], Yingbin Zou [1] and Qiyuan Tang [1]

[1]  College of Agronomy, Hunan Agricultural University, Changsha 410128, China; hbzheng@hunau.edu.cn (H.Z.); yuebao2022@126.com (X.T.); yeshangtianxia@163.com (J.W.); ybzou123@126.com (Y.Z.); qytang@hunau.edu.cn (Q.T.)
[2]  College of Resources and Environment, Hunan Agricultural University, Changsha 410128, China
*  Correspondence: xu7541@hunau.edu.cn; Tel.: +86-731-8461-8076

**Abstract:** No-tillage (NT) management is considered a leading approach for sustaining crop production and improving soil and environmental quality. Based on a long-term no-tillage experiment in a rice–oilseed rape cropping system, we examined differences in soil organic matter (SOM), soil microbial carbon (C) and nitrogen (N) content, and methane ($CH_4$) and carbon dioxide ($CO_2$) fluxes between NT and conventional tillage (CT) management. SOM under NT was 21.0 g kg$^{-1}$, and a significant difference was detected between 2004 and 2016. SOM increased under NT and CT by averages of 0.60 and 0.32 g kg$^{-1}$ year$^{-1}$, respectively. Soil microbial C and N content were higher under CT than under NT. However, soil C:N ratios under NT were 17.4 and 9.7% higher than the CT, respectively, whereas soil microbial C:N ratios under NT were on average 9.47 and 9.70% higher. In addition, about 70% of $CO_2$ net uptake and over 99% of net $CH_4$ emissions occurred during the rice season in May–September in the rice–oilseed rape cropping system. Annual cumulative $CH_4$ and daytime net ecosystem $CO_2$ exchange (NEE) under NT was 1813.9 g $CO_2$ equiv. m$^{-2}$, 10.8% higher than that under CT. Our results suggest that a higher soil microbial C:N ratio and NEE ($CH_4$ and daytime $CO_2$) could contribute to increasing SOM/C in the surface soil under NT management.

**Keywords:** C:N ratio; net ecosystem exchange; soil microbial carbon; soil microbial nitrogen

## 1. Introduction

In recent years, a farming method with low labor requirements and simpler operation, no-tillage (NT), had been more and more widely promoted in modern agricultural production. In China, about 7 million hm$^2$ of cultivated land are under NT management, accounting for 5.5% of the total area under conservation agriculture worldwide [1]; cropland NT management started in 1970, with cereal crops including rice (*Oryza sativa* L.), corn (*Zea mays* L.), wheat (*Triticum aestivum* L.), and economical crops including soybeans (*Glycine max* L.) and oilseed rape (*Brassica campestris* L.). NT management is considered a leading approach to sustaining crop production and addressing soil and environmental quality concerns [1–3].

Studies have found that NT can improve crop yield [4–6]. The reason may be that NT can improve soil water content [6,7]; increase soil nitrogen, phosphorus, and other nutrients [8,9]; reduce soil nutrient leaching [10]; and improve soil enzyme activity [11] and increase microbial biomass [12,13]. The conclusion of NT on improving environmental quality is still controversial. A previous study comparing multiple tillage methods found that NT can reduce $CH_4$ emissions and global warming potential (GWP) of rice fields, but increase $N_2O$ emissions [14]; compared with conventional tillage (CT), NT increases $CH_4$ emissions in winter and decrease $N_2O$ and $CO_2$ emissions in summer [15]; and compared with reduce tillage (RT), it was found that NT can increase SOC and decrease greenhouse gas intensity (GHGI) [16]. However, some studies have shown that no-tillage significantly

increases the $CH_4$ emission from rice fields and enhances the net comprehensive warming potential (GWP) and greenhouse gas intensity (GHGI) of rice fields [17], and significant increase in $N_2O$ emissions [18]. In addition, many studies discussed the impact of NT on carbon (C) sequestration [2] and economy [19].

Previous studies have mainly focused on dryland and dryland crops, and there is also great controversy over whether NT can reduce greenhouse gas emissions, so it is necessary to study the impact of NT management on soil organic carbon (SOC), especially under dryland rice rotation systems. The dynamics of soil microbial carbon, nitrogen and net greenhouse gases can more accurately determine the impact of no-tillage on greenhouse gas emissions. Our hypotheses are that no-tillage increases soil nutrient content and reduces greenhouse gas emissions than conventional tillage under dry-wet rotation. To test our hypothesis, based on a fixed experiment of no-tillage management in a rice–oilseed rape cropping system, we measured $CH_4$ and $CO_2$ fluxes and net ecosystem exchange (NEE) using an ultraportable greenhouse gas analyzer and a static transparent chamber, and measured the changes of SOC and dynamic changes in soil microbial C and N and to evaluate the effect of NT management on $CH_4$ and daytime $CO_2$ NEE.

## 2. Materials and Methods

### 2.1. Experimental Design and Field Management

Starting in 2004, a long-term experiment was conducted based on a rice–oilseed rape cropping system at Changsha (28°11′ N, 113°04′ E), Hunan Province, China. The study site has a moist subtropical monsoon climate with a mean annual temperature of about 17.0 °C, mean annual rainfall of about 1355 mm, and about 1677 h mean annual sunshine [20]. The soil of the experimental field was clayey soil with 1.5% organic matter and 0.14% total N [20]. Liangyoupeijiu, the first super-hybrid rice variety in China, and Xiangzayou 6, a hybrid oilseed rape cultivar, were used in the experiment [21].

In this fixed experiment, a randomized block design was established with four different tillage and cultivation treatments, including conventional tillage (CT) and transplanting (CTTP), NT and transplanting (NTTP), CT and direct seeding (CTDS), and NT and direct seeding (NTDS). Each field plot was 30 × 30 m, with four replicates. More detailed information about the treatments and other management practices are reported in Huang et al. (2011). Based on this experiment, we measured soil greenhouse gas emissions ($CO_2$ and $CH_4$) in the CTTP and NTTP treatments in situ and in real time during 2015–2016. Soil microbe C and N, as well as soil chemical properties in the different layers, were also measured.

### 2.2. Soil Chemical Properties

In 2004, soil samples from the 0–20-cm soil layer were used to determine the soil fundamental fertility; in 2016, soil samples in the 0–60-cm soil profiles were used to determine soil chemical properties. Soil profiles were divided into six layers (0–10, 10–20, 20–30, 30–40, 40–50, and 50–60 cm). Soil samples by the five-point method were dried naturally and then used to determine soil organic matter (SOM) by $K_2Cr_2O_7$-concentrated $H_2SO_4$ and heating. Soil total N was determined by the Kjeldahl method, which involved two steps: (1) digestion of the sample to convert organic N into $NH_4^+$-N and (2) determination of $NH_4^+$-N in the digest. The soil C:N ratio was calculated by dividing the SOC concentration by the total N concentration. Soil total P was conducted using the $H_2SO_4$-$H_2O_2$-HF method and determined using colorimetric method. Soil total K was conducted using micro-diffusion and determined using flame spectrophotometry. Soil total P was determined by molybdenum antimony-D-iso-ascorbic acid colorimetry (MADAC). $NH_4OAc$-extractable K of soil samples was determined by flame spectrophotometry.

### 2.3. Real-Time $CH_4$ and $CO_2$ Flux Measurements

Real-time $CH_4$ and $CO_2$ fluxes were determined using the static chamber method with an ultraportable greenhouse gas analyzer ($CH_4$, $CO_2$, $H_2O$; Los Gatos Research (LGR,

San José, CA, USA). The static chamber was a square box with a side length of 50 cm and height of 120 cm. A fluted base matching that of the static chamber was planted in the soil in advance. Sampling was conducted at 9:00–11:00 a.m. and 15:00–17:00 p.m. on sunny days, and testing within the plot took 5 min. The sampling dates were 20, 37, 57, 77, and 102 days after rice transplanting in the rice season and 127, 159, 187, 242, and 239 days after rice transplanting in the oilseed rape season.

Temperature was recorded accurately in the static chamber and in the 0–3-cm soil layer. Plants were sampled from a 0.24-m$^2$ area within each plot on the sampling date. Plant samples were separated into leaf, straw, and grains by hand, the volume was determined using the drainage method. The drainage method was that plant samples was immersed in the fixed volume vessel (1000, 2000, and 5000 mL), and collected and measured the water volume by other volume vessel. Lastly, the effective volume of the chamber was used to determine the volume of plants in the chamber.

### 2.4. Daytime and Seasonal CO$_2$ and CH$_4$ Net Ecosystem Exchange

Daytime CO$_2$ (F, g m$^{-2}$ d$^{-1}$) and CH$_4$ net ecosystem exchange (F, g m$^{-2}$ d$^{-1}$) were calculated as follows:

$$F = \frac{P \times V}{R \times A \times (T + 273.15)} \times \frac{dc}{dt} \times \frac{M \times S}{10^6} \tag{1}$$

where P is atmospheric pressure under standard conditions ($101.2237 \times 10^3$ Pa); V is the effective volume in the chamber (m$^3$), i.e., the difference between the volume of static chamber and the volume of the plant, fan, and thermometer; R is the gas constant (8.3144 J mol$^{-1}$ K$^{-1}$); A is the covering area of the chamber (m$^2$); T is the average temperature at the testing time inside the chamber (°C); dc/dt is the rate of change in CO$_2$ and CH$_4$ concentrations (mg dm$^{-3}$ s$^{-1}$); M is the CO$_2$ or CH$_4$ relative molecular mass (g mol$^{-1}$); and S is the duration of the sampling day (s).

Seasonal emissions in daytime CO$_2$ and CH$_4$ were calculated as follows:

$$T = \sum [(F_i + F_{i+1})/2 \times d] \tag{2}$$

where T (g m$^{-2}$) is the total seasonal emissions, $F_i$ and $F_{i+1}$ are the measured fluxes on two consecutive sampling days, and d is the number of days between the two sampling dates.

### 2.5. Soil Microbe C and N

In 2016, the five-point method was used to sample soil in the 0–20-cm soil layer. The sampling dates were prior to rice planting, at 20, 37, 57, 77, 102, 119, and 159 days after rice transplanting during the rice season, and 187, 239, 281, and 314 days after rice transplanting in the oilseed rape season. Fresh soil samples were used directly to determine soil microbial C and N using the chloroform fumigation–incubation and K$_2$SO$_4$ extraction methods. Here, soil microbial carbon (mg kg$^{-1}$) = EC/0.33; soil microbial nitrogen (mg kg$^{-1}$) = EN $\times$ 5.0, where the soil microbial C and N coefficients are 0.33 and 5.0, respectively, and EC and EN are the differences in organic C and N in the K$_2$SO$_4$ extraction solution between fumigation and non-fumigation.

### 2.6. Data Analyses

Means of the Indexes were compared using Microsoft Excel 2007 software (Microsoft Corporation, Redmond, WA, USA) and Fisher's least significant difference (LSD) method. We performed a factorial analysis of variance and a least squares difference to test for statistically significant differences between the NT and CT using Statistix 8.0. (Analytical software, Tallahassee, FL, USA).

## 3. Results

### 3.1. Soil Chemical Properties

Under no-tillage (NT) and conventional tillage (CT) management, soil organic carbon (SOC) increased by averages of 0.60 and 0.32 g kg$^{-1}$ year$^{-1}$, respectively (Table 1). In 2016, SOC under NT was 21.0 g kg$^{-1}$, and a significant difference was observed between 2004 and 2016. Soil total N decreased by averages of 19 mg kg$^{-1}$ year$^{-1}$ under NT and 26 mg kg$^{-1}$ year$^{-1}$ under CT. Soil total N and P under NT increased by 5.7 and 5.1%, respectively, compared with those under CT. There was no significant difference between 2004 and 2016. Soil C:N ratios under NT and CT were 17.4 and 15.8, respectively, with that under NT being 9.7% higher than that under NT (Figure 1). Soil total K decrease by an average of 0.51 g kg$^{-1}$ year$^{-1}$ under NT and 0.43 g kg$^{-1}$ year$^{-1}$ under CT; there was a significant difference in these parameters between 2004 and 2016. Soil NaOH hydrolysable N and Olsen P increased across years, and soil NH$_4$OAc extractable K decreased dramatically across years, by 8.28 mg kg$^{-1}$ year$^{-1}$ under NT and 7.51 mg kg$^{-1}$ year$^{-1}$ under CT. Soil chemical properties decreased as soil layer depth increased (Table 2).

**Table 1.** Variation of 0–20 cm soil layer soil chemical properties between the NT and CT treatment from 2004 to 2016.

| Year | SOC (g·kg$^{-1}$) | | STN(g·kg$^{-1}$) | | STP(g·kg$^{-1}$) | | STK(g·kg$^{-1}$) | | SNN(mg·kg$^{-1}$) | | SOP(mg·kg$^{-1}$) | | SNK(mg·kg$^{-1}$) | |
|---|---|---|---|---|---|---|---|---|---|---|---|---|---|---|
| | CT | NT | CT | NT | CT | NT | CT | NT | CT | NT | CT | NT | CT | NT |
| 2004 | 15.0 | | 1.40 | | 1.18 | | 18.10 | | 137.0 | | 38.4 | | 113.0 | |
| 2016 | 18.2 | 21.0 | 1.15 | 1.21 | 1.17 | 1.23 | 13.79 | 12.99 | 154.8 | 154.7 | 52.3 | 57.6 | 37.9 | 30.2 |
| Year | ns | * | ns | ns | ns | ns | * | * | ns | ns | ns | ns | * | * |
| Tillage # | * | | ns | | ns | | ns | | ns | | ns | | ns | |

Soil sample with three replications using the five-point method (*n* = 3). # Significant difference (*p* < 0.05) between 2004 and 2016; SOC, soil organic matter; STN, soil total N; STP, soil total P; STK, soil total K; SNN, soil NaOH hydrolysable N; SOP, soil olsen P; SNK, soil NH$_4$OAc extractable K. * are significantly different according to LSD at *p* < 0.05. ns are not significantly different according to LSD at *p* = 0.05.

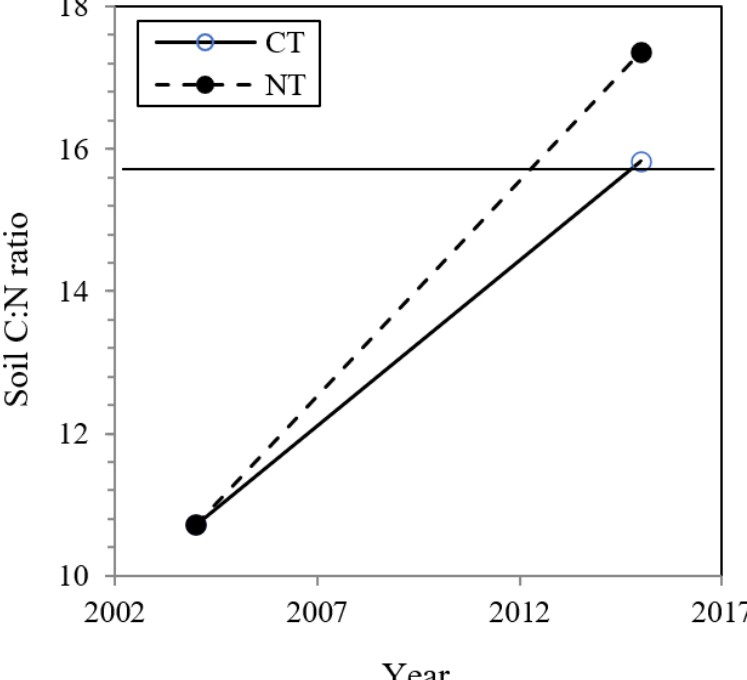

**Figure 1.** Variation in the soil carbon: nitrogen (C:N) ratio under CT and NT management in a rice–oilseed rape cropping system from 2004 to 2016.

**Table 2.** Variation of 0–60 cm soil layer soil chemical properties in 2016.

| Soil Layer (cm) | SOC (g·kg⁻¹) | | STN (g·kg⁻¹) | | STP (g·kg⁻¹) | | STK (g·kg⁻¹) | | SNN (mg·kg⁻¹) | | SOP (mg·kg⁻¹) | | SNK (mg·kg⁻¹) | |
|---|---|---|---|---|---|---|---|---|---|---|---|---|---|---|
| | CT | NT | CT | NT | CT | NT | CT | NT | CT | NT | CT | NT | CT | NT |
| 0~10 | 18.45 | 21.97 | 1.46 | 1.45 | 1.25 | 1.25 | 12.85 | 12.90 | 186.21 | 189.69 | 57.50 | 60.47 | 43.94 | 36.87 |
| 10~20 | 18.00 | 20.10 | 0.83 | 0.97 | 1.09 | 1.21 | 14.72 | 13.08 | 123.31 | 119.64 | 47.14 | 54.77 | 31.80 | 23.55 |
| 20~30 | 16.87 | 17.67 | 0.68 | 0.69 | 0.80 | 0.95 | 15.56 | 14.85 | 97.05 | 105.75 | 28.03 | 40.57 | 19.18 | 19.90 |
| 30~40 | 16.17 | 15.96 | 0.86 | 0.74 | 0.62 | 0.69 | 16.08 | 16.38 | 104.47 | 95.33 | 17.79 | 25.67 | 20.80 | 20.17 |
| 40~50 | 13.94 | 14.02 | 0.81 | 0.87 | 0.55 | 0.52 | 14.63 | 15.81 | 84.14 | 75.54 | 13.48 | 12.54 | 16.70 | 19.13 |
| 50~60 | 9.38 | 9.47 | 0.65 | 0.62 | 0.39 | 0.44 * | 17.89 | 20.67 * | 60.50 | 70.70 ** | 4.42 | 6.07 * | 17.21 | 19.22 |

SOC, soil organic matter; STN, soil total N; STP, soil total P; STK, soil total K; SNN, soil NaOH hydrolysable N; SOP, soil olsen P; SNK, soil NH$_4$OAc extractable K. *, ** are significantly different according to LSD at $p$ = 0.05 and 0.01 between the NT and CT, otherwise, there was no significant difference between the NT and CT.

### 3.2. Daytime and Annual CH$_4$ and CO$_2$ Net Ecosystem Exchange

Daytime CO$_2$ net ecosystem exchange (NEE) was 14.5% higher in the rice season and 5.9% higher in the oilseed season under NT than under CT (Figure 2a). During the rice season, the maximum daytime CO$_2$ NEE values under NT and CT were 37.28 g m⁻² d⁻¹ and 34.82 g m⁻² d⁻¹ at 57 days after rice transplanting. During the oilseed season, daytime CO$_2$ NEE was highest from 222 to 316 days after rice transplanting. There was a dramatic difference in daytime CO$_2$ NEE between the rice and oilseed seasons (Figure 3). Daytime CO$_2$ NEE during the rice season was 70.0% under NT and 68.4% under CT, whereas that during the oilseed season was 30.0% under NT and 21.6% under CT.

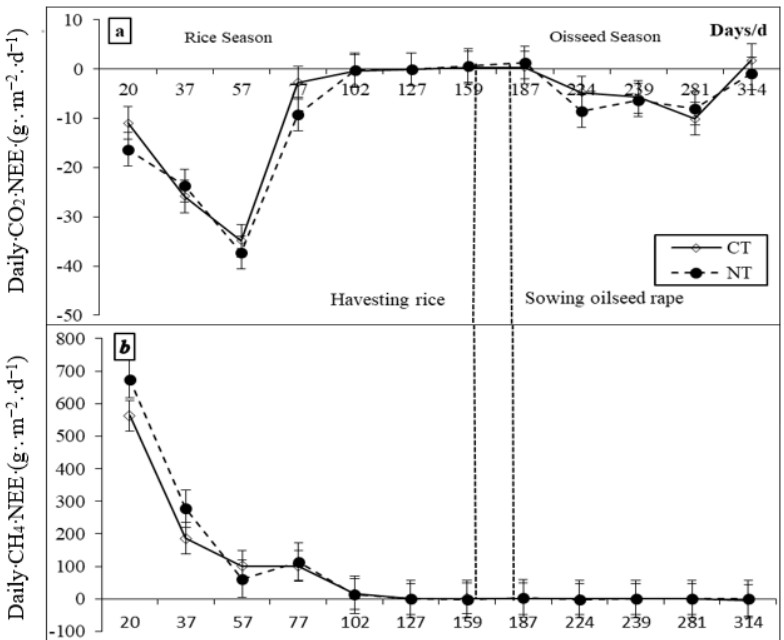

**Figure 2.** Changes in daily CO$_2$ (**a**) and CH$_4$ (**b**) net ecosystem exchange (NEE) under conventional tillage (CT) and no-tillage (NT) management in a rice–oilseed rape cropping system. The bars mean the standard error.

The ranges of CH$_4$ NEE were –0.38 to 674.70 mg m⁻² d⁻¹ during the rice season and –4.28 to 3.29 mg m⁻² d⁻¹ during the oilseed season (Figure 2b). CH$_4$ NEE during the rice season was 15.0 g m⁻² season⁻¹ under NT, accounting for 99.6% of total CH$_4$ NEE. CH$_4$ NEE during the rice season was 12.9 g m⁻² season⁻¹ under CT, 16.6% lower than that under NT in the rice–oilseed rape cropping system.

Total CH$_4$ emissions were 376.9 g CO$_2$ equiv. m⁻² under NT and 323.4 g CO$_2$ equiv. m⁻² under CT. In the rice–oilseed rape cropping system, annual CH$_4$ and daytime CO$_2$ NEE under NT was 1813.9 g CO$_2$ equiv. m⁻², 10.8% higher than that under CT.

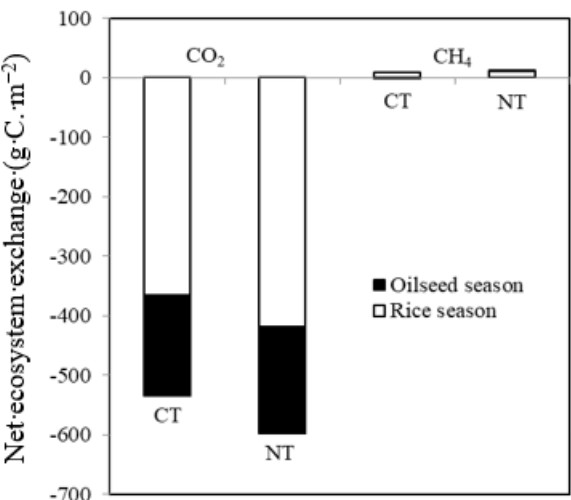

**Figure 3.** Seasonal $CH_4$ and $CO_2$ net ecosystem exchange (NEE) under CT and NT management in a rice-oilseed rape cropping system.

### 3.3. Soil Microbial C and N

The annual variation in soil microbial carbon (C) and nitrogen (N) was consistent between NT and CT management (Figure 4a,b). Soil microbial C and N under CT were higher by averages of 16.0 and 32.9%, respectively, than those under NT, whereas the soil microbial C:N ratio under CT was lower by an average of 9.7% than that under NT (Figure 4c).

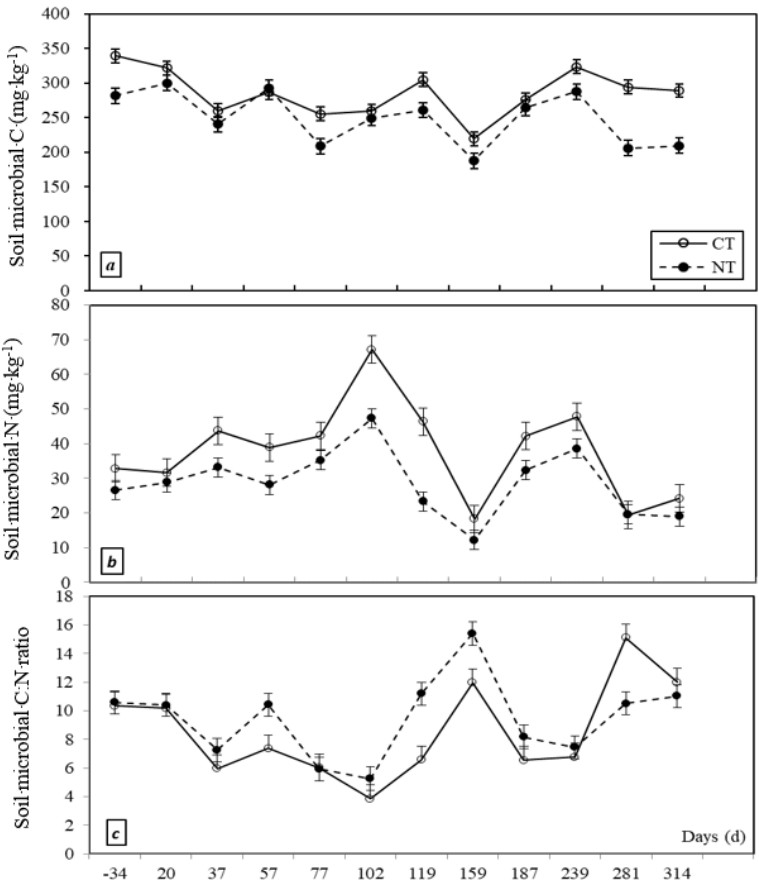

**Figure 4.** Variation in soil microbial C (**a**), N (**b**), and C:N (**c**) ratio under CT and NT management in a rice-oilseed rape cropping system in 2015–2016. The bars mean the standard error.

## 4. Discussion

Rice fields are a complex agro-ecological area, and greenhouse gases and soil fertility will be constantly changing and changing by various activities, in which farming methods can directly change the soil. Yonemura et al. (2014) reported that no-tillage (NT) management was generally effective in mitigating total global warming potential through reduced soil respiration and $N_2O$ emission in temperate regions [22]; Li et al. (2014) found that NT can reduce soil disturbance, increase the stability of aggregates, and facilitate the formation of refractory carbon, thereby reducing soil $CO_2$ emissions [23]. Compared with conventional tillage (CT), NT can also enhance $CH_4$ oxidative and methanotrophic activity, ultimately reducing $CH_4$ emissions. In this study, we found that NT did not decrease the emission of $CO_2$ and $CH_4$. During the growth of crops, the absorption of $CO_2$ and the release of $CH_4$ by NT were higher than those of CT. This may be mainly due to the slow entry of nitrogen fertilizer into the soil under NT and the higher C:N ratio, which weakened the microbial activity and the ability to compete for nutrition, resulting in the weakening of the decomposition of $CH_4$ and the weakening of the competition for microbial nutrition for crops. On the contrary, the growth was better than that of CT. Finally, it may increase the $CO_2$ absorption in the daytime and the $CO_2$ emission at night.

The soil carbon:nitrogen (C:N) ratio is an important soil fertility indicator due to the close interactive relationship between soil organic carbon (SOC) and total N [24]. Wan et al. (2015) reported that the soil C:N ratio was an important factor influencing soil microbial community structure in subtropical coniferous and broadleaf forest plantations [25]. Under NT management, positive changes in soil physical properties appear to be closely related to positive effects of NT on SOC accumulation [7]. Our study found that although the soil microbial carbon and soil microbial nitrogen of CT were slightly higher than those of NT, which may be caused by CT directly changing the soil structure and making it easier for nutrients to enter the soil, but the soil microbial C:N ratio of CT was lower than that of NT, indicating that the ability of microorganisms to compete for fat may be reduced under NT. Therefore, further studies are needed to explain fully the phenomenon.

Previous studies have reported that the NT management can increase SOC with soil physical structure [26], fertilizer N input [27], and crop rotation [28], and that cropping frequency and fertilizer N input in association with NT resulted in increased SOC [29], and these just explain why NT under crop rotation and fertilization in this study could increase the SOC content in soil more than CT. Xiao et al. (2020) found that long-term NT was more beneficial to SOC increase in soil surface [30]. Many other studies have reported that NT increases SOC only in the upper 10 cm of the soil [2,31]. Even in long-term (>30 years) tillage studies, NT appears to increase wet aggregate stability only in the upper 10 cm of the soil [13,31], this is consistent with our research results. In the rice–rapeseed planting system, reasonable fertilizer nitrogen input and NT treatment can significantly increase the SOC content in the upper soil layer, especially in the 0-10 cm soil layer.

In addition to studying soil physical properties (reviewed by Blanco-Canqui and Ruis, 2018), chemical properties [32], and green-house gas (GHG) emissions, there are many reports that have looked at the effects on GHG emissions from other perspectives. For example, Kulmány et al. (2022) reported that moisture content, air temperature and pressure all play important roles in $CO_2$ emissions in NT systems [33]; and Yonemura et al. (2014) report that $CH_4$ emissions increased significantly under NT in a wet, temperate climate [22]. Therefore, in addition to soil physical properties, chemical properties and microbial activities, greenhouse gas emissions or absorption under NT management are also affected by climate, temperature, and other external factors. Most of the current studies have studied the impact of NT on greenhouse gases by measuring several related indicators, which is too one-sided and subjective, resulting in great differences in research conclusions. How to more comprehensively explore the impact of tillage methods on greenhouse gas emissions is worth pondering. Such as through the actual field combined with computer model comprehensive study of NT is how to affect greenhouse gas production [34].

## 5. Conclusions

In a fixed experiment in a moist subtropical monsoon climate, we found that soil organic matter (SOM) was 21.0 g kg$^{-1}$ under no-tillage (NT), with a significant difference observed between 2004 and 2016. Under NT and conventional tillage (CT), SOM increased by averages of 0.60 and 0.32 g kg$^{-1}$ year$^{-1}$, respectively. Soil microbial carbon (C) and nitrogen (N) under CT were higher than those under NT; however, the soil C:N ratio was 17.4 under NT, 9.7% higher than that under CT, and the soil microbial C:N ratio under NT was an average of 9.47, 9.7% higher than under CT on average. Using an ultraportable greenhouse gas analyzer and static transparent chamber method of circulation gas recovery, we found that about 70% of daytime $CO_2$ net uptake and over 99% of $CH_4$ net emission occurred during the rice season in a rice–oilseed rape cropping system, and annual $CH_4$ and daytime $CO_2$ net ecosystem exchange under NT was 1813.9 g $CO_2$ equiv. m$^{-2}$, 10.8% higher than that under CT. Consequently, long-term no-tillage could increase soil organic matter/carbon in the surface soil, and increase annual net ecosystem exchange in a rice–oilseed rape cropping system, especially $CH_4$ emission; more attention is needed on to how to reduce $CH_4$ emission under the background of climate change.

**Author Contributions:** Conceptualization, H.Z., H.X. and Y.Z.; methodology, H.Z.; validation, H.Z. and X.T.; investigation, H.Z. and J.W.; writing—original draft preparation, H.Z.; writing—review and editing, H.Z. and H.X.; supervision, Y.Z.; project administration, H.X. and Q.T.; funding acquisition, Q.T. All authors have read and agreed to the published version of the manuscript.

**Funding:** This research was fund by the Earmarked Fund for China Agriculture Research System (Grand No. CARS-01-27).

**Institutional Review Board Statement:** Not applicable.

**Informed Consent Statement:** Not applicable.

**Data Availability Statement:** The datasets generated during and/or analyzed during the current study are available from the corresponding author on reasonable request.

**Acknowledgments:** We are thankful to anonymous reviewers and editors for their helpful comments and suggestions.

**Conflicts of Interest:** The authors declare that they have no competing interests.

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
