# Peer review of "Effect of No-Tillage Management on Soil Organic Matter and Net Greenhouse Gas Fluxes in a Rice-Oilseed Rape Cropping System"

_agriculture, doi:10.3390/agriculture12070918_

Round 1

Reviewer 1 Report

Dear Authors and Editors,

the manuscript has been considerably improved according my suggestions. I read it and now its reading is correct, and I don't detect any mistakes.

Best regards

Author Response

Dear the reviewer,thank for your time and attetion.

Reviewer 2 Report

The authors have answered most of my doubts and comments from the previous review, for which I thank them. 

I still have some minor comments on the new version:

Line 129: Specify the units of the rate of change dc/dt

Line 155: carbon

Line 203: Specify what the bars mean (standard error?)

Lines 290 - 293: Improve the wording of this paragraph

Author Response

Dear the reviewer:

Firstly, thank for your time and attention.

The authors have answered most of my doubts and comments from the previous review, for which I thank them. 

I still have some minor comments on the new version:

Line 129: Specify the units of the rate of change dc/dt

Answer: The units was mg m-3 s-1.

Line 155: carbon

Answer: we have revised the word in the revised manuscript.

Line 203: Specify what the bars mean (standard error?)

Answer:  The bars means the standerd error.

Lines 290 - 293: Improve the wording of this paragraph

Answer:  we have been revised in the revised manuscript.

Lastly, Thank for your time again.

Reviewer 3 Report

The article is well revised in the light of reviewer comments. 

Author Response

Dear the reviewer, thank for your time and attention.

This manuscript is a resubmission of an earlier submission. The following is a list of the peer review reports and author responses from that submission.

Round 1

Reviewer 1 Report

Dear Editor of Agriculture,

Dear Authors,

thank you for giving me the opportunity to review the manuscript n. agriculture-1677476 entitled “Effect of no-tillage management on soil organic matter and net greenhouse gas fluxes in a rice–oilseed rape cropping system”.

The manuscript aims to the evaluation of greenhouse gases emission, in particular CO2 and CH4, in a rice -oilseed rape rotation with the context of a long term experiment. Other soil measurements, such as SOM, microbial carbon and soil nitrogen content were measured in the experiments. The topic is in line with the journal scope. However, some aspects should be addressed.

In all sections of the manuscript the references are not reported correctly, therefore they should be formatted according with the “guide of authors” reported in the Agriculture journal homepage.

INTRODUCTION

Line 25-26 I suggest eliminating this sentence, it is not of interest for the scope of the journal.

Line 34 in the reference there is a semicolon that should be removed.

In the introduction is missing the hypothesizes, I suggest to add in order to add important information for the readers regarding what the manuscript aims with the research.

MATERIALS and METHODS

Line 67 only NT treatment is applied in the long term experiment? As reported in the experimental design there is also tillage treatment and the term “NT” in the sentence could generate confusion for the readers.

The data analysis should be better described. Please rewrite the subsection.

RESULTS

Table 2 report the value but there is not clear information regarding the significance of the data, in my opinion the data should be managed differently by reporting symbols that state the differences or not regarding the value reported for the treatments.

Figure 2 needs to be improved, unfortunately the lines cover the values reported in the axis. However, what is not clear, from my point of view is the negative value on CO2, I suppose this is the variation between two consecutive measurements but it is not reported in any section of the manuscript. I suggest to the author to make a strong revision of the materials and methods section reporting detailed information on data management and how they are reported in the results. I also suggest to add vertical bar to each point reporting the standard error (SE) in order to correctly evaluate the information. Similarly, the results reported in figure 3, is not clear why the report a negative values. In addition there is not mention if the data reported are significative different or no. 

DISCUSSION

This section should focus mainly on the obtained results, at this time it is very general and do not leave a clear message to the reader concerning the significance of the work done.

Based on the above consideration, I would ask to the author to improve the manuscript and send the revised version focusing on the aspects cited.

Reviewer 2 Report

  1. The abbreviations in the Abstract and all across the draft need to be elaborated the first time they appear.
  2. The abstract language is very poor. Re-write the abstract with concise information delivery of tangible results.
  3. The introduction section is very general. Hypothesis should be clearly defined.
  4. The abbreviations in the results & discussion and all across the draft need to be elaborated the first time they appear.
  5. Please use only SI units in the results. Please try to induct latest references and keep the number of references to below 50. 
  6. Table & Fig. titles need appropriate headings.
  7. The discussion needs induction of logical reasoning’s with latest references, which is quite lacking in the draft….Improve it.
  8. In conclusion, the concluding statement and its application is lacking...add it.

Reviewer 3 Report

The manuscript titledEffect of no-tillage management on soil organic matter and net greenhouse gas fluxes in a rice–oilseed rape cropping system describes the change in soil organic matter content in a long-term experiment on conservation tillage, and incorporates measurements during a growing season of a rice-oilseed rape crop rotation.

Specific comment and suggestions to the authors are included in order to improve the final version of the manuscript.

 General comments:

- The results obtained can only be considered preliminary because only one measurement season is too short to draw any conclusions...

- How do the authors determine the net ecosystem exchange (NEE)? The material and methods section does not explain how the net exchange of the ecosystem is determined.

- Analytical methods are not referenced.

- Attached are some references that may be of interest to the authors in order to introduce and discuss their results of gaseous emissions.

Abdalla, M., Osborne, B., Lanigan, G., Forristal, D., Williams, M., Smith, P., & Jones, M. B. (2013). Conservation tillage systems: a review of its consequences for greenhouse gas emissions. Soil use and management, 29(2), 199-209.

Huang, Y., Ren, W., Wang, L., Hui, D., Grove, J. H., Yang, X., ... & Goff, B. (2018). Greenhouse gas emissions and crop yield in no-tillage systems: A meta-analysis. Agriculture, Ecosystems & Environment, 268, 144-153.

Mangalassery, S., Sjögersten, S., Sparkes, D. L., Sturrock, C. J., Craigon, J., & Mooney, S. J. (2014). To what extent can zero tillage lead to a reduction in greenhouse gas emissions from temperate soils?. Scientific reports, 4(1), 1-8.

Malhi, S. S., Lemke, R., Wang, Z. H., & Chhabra, B. S. (2006). Tillage, nitrogen and crop residue effects on crop yield, nutrient uptake, soil quality, and greenhouse gas emissions. Soil and Tillage Research, 90(1-2), 171-183.

Specific remarks:

- Line 15: this sentence is incorrect:

…soil C/N ratios under NT averaged 17.4, 9.7% higher than the CT treatment.

- Line 17: net uptake? Why extractions, if what is measured are emissions?

- Line 34: (; Palm et al.,… change to (Palm et al.,…

- Line 43: missing subscript CH4

- Line 71: Tidal clay? It would be convenient to describe the type of soil according to one of the accepted soil classifications.

- Line 88: Specify the number of soil samples taken.

- Line 91: correct ammonium subscripts and superscripts.

- Lines 94: “Soil total P was determined…” is incorrect. Do you mean P available?

It would be convenient to include in this section the initials used in Table 1 to refer to each of the elements determined (STN, STP,...).

- Line 109: drainage method? Some reference or a brief description of the method would be helpful.

- Lines 110 – 112: It is not clear for what purpose LAI and aboveground biomass are determined.

- Line 120: 8.3144 J mol-1 K-1

- Lines 127 – 128: it is assumed that the variation between two days of measurement is linear, but this need not be the case, depending on weather conditions. Some comment on the possible error made with this assumption would be helpful.

- Line 130: How many samples? Were several points sampled and then a composite sample was made?

- Line 157: It would be convenient to indicate the number of samples (n=xx).

- Line 166: specify that these are differences between treatments and not between soil depths.

- Lines 168 – 169: Figure 2a shows the CO2 emission fluxes but not the NEE.

- Line 170: The values do not correspond to what is shown in the figure.

- Line 186: Are they average values of the two measurement periods on each day? If they are average values of four replicates, why are no error bars or standard deviation included?

- Lines 222 – 224. “…the soil microbial C:N ratio of CT was higher than that of NT…” This statement is not correct. The authors state in line 193 the following: “…the soil microbial C:N ratio 193 under CT was lower by an average of 9.7% than that under NT …”. Please clarify. In addition, the following paragraph is not understood.... rewrite.

- Lines 225 – 226: perhaps for this reason, a single season of measurement is too short to draw conclusions...

- Line 231: oxidative

- Line 233: absorption?   But wasn't emission being measured?

- Lines 236 – 238: this is not very well understood... also, where is the growth data?
